

# Engrailed-2 promotes a malignant phenotype of esophageal squamous cell carcinoma through upregulating the expression of pro-oncogenic genes

Yong Cao, Xiaoyan Wang, Li Tang, Yan Li, Xueqin Song, Xu Liu, Mingying Li, Feng Chen and Haisu Wan

Experimental Medicine Center, The Affiliated Hospital of Southwest Medical University, Luzhou, Sichuan, China

## ABSTRACT

**Background:** A number of homeobox genes have been implicated in the development of various cancers. However, the role of engrailed 2 (EN2), a member of the homeobox gene superfamily, in esophageal squamous cell carcinoma (ESCC) remains unknown.

**Methods:** The expression of EN2 was examined using quantitative real-time PCR and immunohistochemistry. A stable cell line was established to express exogenous EN2 using a lentivirus system. The malignant phenotype was analyzed with proliferation, clonogenicity, wound-healing and invasion assays. The CRISPR/Cas9 system was adopted to deplete endogenous EN2. RNA profiling was performed using gene expression microarray. The ShRNA-mediated method was used to knock down the expression of SPARC. The structure-function relationship was determined using site-directed mutagenesis.

**Results:** EN2 is highly expressed in ESCC. The malignant phenotype of the ESCC cell line was amplified by an overexpression of EN2 but was attenuated by a disruption of EN2. RNA profiling analysis revealed that distinct sets of genes were modulated by the expression of EN2 in various ESCC cell lines and oncogenes were among these. EN2 greatly increased the expression of SPARC in Eca109. Site-directed mutagenesis revealed that the induction of SPARC was closely correlated with the protumor function of EN2. ShRNA-mediated knockdown of SPARC attenuated the malignant phenotype of EN2-infected cells. These data suggest that SPARC is crucial for mediating the protumor function of EN2.

**Discussion:** EN2 has an oncogenic function in ESCC that is mediated by upregulating the expression of pro-oncogenic genes downstream. EN2 may potentially act as a diagnostic marker or therapeutic target for ESCC treatment in the future.

Corresponding author
Haisu Wan, whssyzx@swmu.edu.cn

## INTRODUCTION

Esophageal squamous cell carcinoma (ESCC) is one of the most common malignancies diagnosed in China and worldwide and is associated with high rates of morbidity and

mortality (*Lin et al., 2013*). The overall 5 year survival rate of ESCC patients is in the low range of 15–25% (*Domper Arnal, Ferrandez Arenas & Lanas Arbeloa, 2015*). Significant efforts have been made to determine the molecular events that occur during cancer development with the expectation that a better understanding of the molecular mechanisms will help to identify more diagnostic markers or therapeutic targets and may improve clinical outcomes.

Homeobox genes in humans are part of a superfamily with approximately 300 members. The associated genes have a homeobox region, approximately 180 base pairs in length, which encodes a highly conserved homeodomain (HD) with a length of about 60 amino acids (*Madissoon et al., 2016*). The homeobox proteins are transcription factors that can activate or repress the genes downstream (*Carnesecchi, Pinto & Lohmann, 2018*). The homeodomain is responsible for the sequence-specific binding of the homeobox protein to its target DNA. The three-dimensional structure of the homeodomain consists of three alpha-helices connected by two short loops, which form the structural basis for direct contact with DNA (*Fraenkel et al., 1998*; *Pabo & Sauer, 1992*). Homeobox genes are crucial for normal embryonic development and the maintenance of adult cells (*Friedrich et al., 2016*; *Hueber & Lohmann, 2008*). The aberrant expression of homeobox genes has been observed in a variety of cancers (*Mansour & Senga, 2017*; *Teo et al., 2016*; *Zhang, Li & Zhang, 2018*).

Engrailed 2 is one member of the homeodomain-containing transcription factor. It is engaged in the development of the nervous system at the embryonic stage and in the establishment and maintenance of the spatial integrity of specific regions of developing tissues (*McGrath et al., 2013*; *Morgan, 2006*). Previous studies have demonstrated that the EN2 gene is highly expressed in cancers of the prostate and breast (*Bose, Bullard & Donald, 2008*; *Martin et al., 2005*). Our study provides evidence for the role of EN2 in the development of ESCC by its activation of downstream oncogenic genes.

## MATERIALS AND METHODS

### Cell lines and culture

The ESCC cell lines, Eca109, Kyse150 and TE-1 were obtained from the Type Culture Collection of the Chinese Academy of Sciences (Shanghai, China). The immortalized human esophageal epithelial cell line (HEEC) and 293FT cell lines were obtained from Shanghai Tongpai Biotechnology Co. LTD. (Shanghai, China). HEEC and 293FT cells were maintained in Dulbecco's Modified Eagle's Medium (DMEM) (Invitrogen, Carlsbad, CA, USA) and were supplemented with 10% fetal bovine serum (FBS) (Invitrogen, Carlsbad, CA, USA). The ESCC cell lines were cultured in Roswell Park Memorial Institute 1,640 medium containing 10% fetal bovine serum. All cells were maintained at 37 °C in a 5% CO2-humidified incubator.

### Patients and samples

We collected 32 pairs of ESCC tissues and their matched adjacent tissues from the Department of Thoracic Surgery of the Affiliated Hospital of Southwest Medical University (Lu Zhou, Luzhou City, China). All patients underwent potentially curative

surgery without preoperative chemotherapy or radiotherapy. This study was approved by the ethics review board at the Affiliated Hospital of Southwest Medical University (K2018002-R). All of the participants involved in our study provided written informed consent.

## Total RNA extraction and RT-PCR assay

Total RNA was extracted from tumor samples or cells using Trizol reagent (Invitrogen, Carlsbad, CA, USA). Four hundred nanograms of total RNA from each sample were reverse-transcribed into cDNA using the PrimeScript™ RT reagent kit with gDNA Eraser (Takara, Dalian, China). qRT-PCR was performed using the SYBR Premix Ex Taq II Kit (Takara, Dalian, China) according to the manufacturer's instructions. The specific primers used are shown in Table S1. The expression of GAPDH was used as an internal control. The RNA expression levels of genes were evaluated by the $2^{-\Delta\Delta Ct}$ method. All specimens were examined in triplicate.

## Construction of the lentiviral vectors and establishment of stable cell lines

The plasmids pVAX-EN2 and pCDH-NEO were constructed in the laboratory and used to generate the EN2-expressing vector. Human EN2 cDNA (G136030; YouBio, Hunan, China) was amplified and cloned into pVAX1 for pVAX-EN2 (VT1048; YouBio, Hunan, China) using the HindIII and XhoI sites. The copGFP sequence was replaced with the neomycin gene in the plasmid pCDH-CMV-MCS-EF1-copGFP (JiRan, Shanghai, China) for pCDH-NEO. The EN2 cDNA was then amplified from pVAX-EN2 and subcloned into the XhaI and BamHI sites of pCDH-NEO. The resultant vector was named pCDH-EN2. The ShRNA vector for the human SPARC gene was created by inserting the shRNA-SPARC sequence into the BamHI and EcoRI sites of pLVX-shRNA2 (JiRan, Shanghai, China). A scramble sequence was inserted into the same sites to establish a control vector. All of the constructed vectors were verified by Sanger sequencing.

The related sequences are as follows:

ShRNA-SPARC sense: 5′-gatcc cggcg gttgt tcttt cctca cattt caaga gaatg tgagg aaaga acaac cgttt ttt-3′ and antisense: 5′-aatta aaaaa cggtt gttct tcct cacat tctct tgaat gtgag gaaag aacaa ccgcc gg-3′; scramble sense: 5′-gatcc cggta caaca gccac aacgt ctatt caaga gatag acgtt gtggc tgttg tattt ttt-3′ and antisense: 5′-aatta aaaaa tacaa cagcc acaac gtcta tctct tgata gacgt tgtgg ctgtt gtacc gg-3′.

The stable cell lines were established by cotransfecting 293FT cells with the correct respective lentiviral vector, which was either pCDH-EN2 or pCDH-NEO (control); and the pSPAX2 (VT1444; YouBio, Hunan, China) and pMD2.G (VT1443; YouBio, Hunan, China) package plasmids using PEI reagent (Polysciences, Warrington, PA, USA), following standard laboratory protocol (*Toledo et al., 2009*). The supernatant containing the virus particles was collected after 3 days and was used to infect the Eca109 and Kyse150 cell lines. The positive cell clones were selected using G418 (500 μg/mL; Thermo Fisher Scientific, Waltham, MA USA) and were named EN2-Eca109, NEO-Eca109, EN2-Kyse150 and NEO-Kyse150. To assess the function of SPARC, shRNA-SPARC was lentivirally

infected into either EN2-Eca109 or NEO-Eca109. The positive cells were selected after 48 h using flow cytometry based on the expression of EGFP.

## Immunofluorescent staining

Cells were incubated for monolayer growth on glass slides at 37 °C, 5% $CO_2$ for 24 h. The cell lines were removed from the culture medium and fixed in 4% paraformaldehyde for 15 min at room temperature, followed by an overnight incubation with polyclonal goat anti-EN2 antibody (ab45867; Abcam, Cambridge, UK) diluted (1:100) in 1% BSA. FITC-conjugated rabbit anti-goat immunoglobulin G (ZSGB-Bio, Beijing, China) was used as the secondary antibody (diluted at 1:200). Cells stained with the secondary antibody were used as the negative control group. Images were collected using a Leica confocal microscope (SP8; Leica, Wetzlar, Germany).

## CRISPR construct and clone selection

The DNA sequence surrounding the initial amino acid code in the EN2 locus was selected as the target of this experiment following the guidelines provided on the website of the Feng Zhang Laboratory (http://crispr.mit.edu/). The Cas9 targeting sequence (Cas9-EN2) was formed by annealing two complementary DNA oligos, that is, sgRNA-EN2-sense and sgRNA-EN2-antisense; this sequence was cloned into the BbsI site of the PX458 vector (#48138; Addgene, Watertown, MA, USA). The resultant vector, pCas9-EN2, or PX458 vector (control), was transfected into TE-1 cells using Lipofectamine 3000 (Thermo Fisher Scientific, Waltham, MA, USA) according to the manufacturer's instructions. EGFP-positive cells were selected after 24 h by flow cytometry and were cultured in 96-well plates with a concentration of one cell per well. Individual colonies were propagated after a 2 week incubation period. The genomic DNA was extracted and PCR-amplified using primer-1 and primer-2. The disruption of the gene was further verified using the mismatch-sensitive T7 endonuclease assay (Bressan et al., 2017; Cong & Zhang, 2015).

The sequences of the oligos and primers used in this experiment were as follows: SgRNA-EN2-sense: 5′-caccg cggta gcagc ccggg cgaag-3′; sgRNA-EN2-antisense: 5′-aaacc ttcgc ccggg ctgct accgc-3′; primer-1: 5′-tgtgc aaaga tccga gctgt cagag a-3′; and primer-2: 5′-aacgg gtgga caggg tctct acct-3′.

## Site-directed mutagenesis

Mutants of the human EN2 gene were generated using our own previously established protocol (Wan et al., 2012). pVAX-EN2 was used as the template. The vector sequence was amplified using a pair of vector primers, VP1 and VP2, which were subjected to Esp 3I digestion. The digested product was self-ligated to generate the expected mutant. The vectors with the expected mutations were verified using Sanger sequencing. The mutated EN2 sequences were then cloned into pCDH-NEO. The vector primers and oligos used in this experiment were as follows: Mut.1-VP1: 5′-aatta cgtct cagag ccaga acaag cgcgc caaga t-3′; Mut.1-VP2: 5′-aatta cgtct cagct cagct cctgc gccag gc-3′; Mut.2-VP1: 5′-aattt cgtct caacg ggcaa caaga acacg ctggc-3′; and Mut.2-VP2: 5′-aattt cgtct caccg tgaac caaat cttga tctgt gactc gt-3′.

## Proliferation and colony-formation assay

Cell proliferation was examined using both the Cell Counting Kit-8 assay and the live-cell imaging assay. Cells were seeded into 96-well plates at concentrations of 3,000 cells/well using the Cell Counting Kit-8 (CK04; Dojindo, Tokyo, Japan). The culture medium for each cell was replaced at 24, 48 and 72 h with a solution containing 100 µL of fresh medium and 10 µL of CCK-8 solution; the cells were incubated at 37 °C in the dark for 1 h. The absorbance (A) value of each well was measured at a wavelength of 450 nm. Cells were seeded in 96-well culture plates for the live-cell image assay at concentrations of 3,000 cells/well and were placed in the IncuCyte Live-Cell Imaging System (Essen Bioscience, Ann Arbor, MI, USA). Growth curves were calculated from confluence measurements using image analysis software. Cells were seeded in six-well plates for the colony formation assay at concentrations of 200 cells/well and the growth medium was changed at 3 day intervals. The clones were fixed with 4% formaldehyde for 10 min and stained with freshly prepared diluted Giemsa solution for 15 min. The clones were then counted and photographed.

## Wound healing and matrigel invasion assay

Cells were plated in 96-well plates. After the cells reached 90–100% confluency, a wound was made using the Essen Bioscience 96-pin wound maker (Essen BioScience, Ann Arbor, MI, USA), which generated a cell-free space approximately 600 µm wide. The cells were washed twice with PBS to remove the floating cells. A total of 100 µL of culture medium containing 1% FBS was added to the cells for a migration assay. A Matrigel invasion assay was performed in which cells were covered with 50 µl of Matrigel solution containing 3 mg/mL Matrigel (Matrigel Basement Membrane Matrix, 354234; BD Biosciences, San Jose, CA, USA) that was solidified after an incubation period of 30 min at 37 °C. The matrix was overlaid with an additional 100 µL of normal growth medium. Wound confluency was monitored using the IncuCyte Live-Cell Imaging System. A cell wound healing software module (Essen BioScience, Ann Arbor, MI, USA) (*Gujral et al., 2014*) was used for image analysis. The wound width was used to calculate the rate of migration and invasion.

## Gene expression profiling

Total RNA was extracted using Trizol (Invitrogen, Carlsbad, CA, USA); a total of 500 ng of RNA was used for cDNA synthesis. The following experiment was performed in consultation with Gene Company Ltd. (Hong Kong, China). Gene expression profiles were accessed using the human Affymetrix Clariom S Assay (902926). Microarrays were scanned using the GeneChip® Scanner 3000 7G. The data were analyzed with the robust multichip analysis (RMA) algorithm using default analysis settings and global scaling. Normalization method values were presented according to log2 RMA signal intensity. R package limma (*Ritchie et al., 2015*) was used to obtain differential expression genes (DEGs); the cutoff fold-change was set at >1.5 or <−1.5 and $p$-value < 0.05. The DEGs

were applied in Gene Ontology analysis by R package ClusterProfiler (*Yu et al., 2012*). The transcriptomic data in this article has been uploaded to the GEO repository with the Accession Number GSE136331.

## Western blot assay

The total cellular protein was extracted using a pre-cooled RIPA buffer (Cell Signaling, Danvers, MA, USA) with protease and phosphatase inhibitors. Protein samples were separated by SDS-PAGE and then transferred to 0.45 μm PVDF membranes (Millipore, Billerica, MA, USA). The membranes were soaked in a blocking solution of 5% skim milk in Tris-buffered saline (TBS) for 1 h and incubated overnight at 4 °C with a primary anti-SPARC antibody (AF941, R&D), a primary antibody dilution buffer (P0023A, Beyotime) diluted to a ratio of 1:1,000, and anti-β-tubulin antibody (AF1216; R&D) with a 1:1,000 dilution. The blots were subsequently incubated with the homologous HRP-conjugated secondary antibody (Sigma–Aldrich, St. Louis, MO, USA). Immunoreactive bands were detected using ECL reagents (Thermo Fisher Scientific, Waltham, MA, USA).

## Immunohistochemistry

Formalin-fixed, paraffin-embedded sections were treated with 3% $H_2O_2$ for 10 min after routine deparaffinization in xylene and rehydration in decreasing concentrations of ethanol (100%, 95%, 85% and 75%), followed by a period of heating in citrate sodium for antigen retrieval for approximately 3 min. The sections were incubated with 10% normal goat serum for 20 min to block nonspecific reactions after antigen retrieval. The sections were then incubated overnight at 4 °C with mouse monoclonal antibody, diluted at 1:200, against human EN2 (Santa Cruz Biotechnology, Dallas, TX, USA). After the incubation of the primary antibody, the streptavidin/peroxidase amplification kit (ZSGBBio, Beijing, China) was used for the EN2 antigen–antibody reaction. The sections were treated with diaminobenzidine to promote the appearance of the EN2 signal. Two pathologists were invited to independently score the immunohistochemical signals according to intensity (0–3) and positive percentage (0–100%). The intensity of positive staining was scored as: 0, negative (−); 1, weak (+); 2, moderate (++); and 3, strong (+++). The expression of EN2 was calculated as the product of the intensity and positive percentage × 100 (IHC score).

## Statistical analysis

SPSS version 16.0 software (IBM Corp., Armonk, NY, USA) was used for statistical analysis. All experimental data are presented as the mean ± standard deviation from at least three separate experiments. The significance of the mean values between the two groups was analyzed using a two-tailed Student's *t*-test. Pearson's correlation coefficient was used to analyze the mRNA expression data obtained from qRT-PCR to provide a correlation analysis of EN2 and SPARC expression in the ESCC tissue samples. Statistical *p*-values of less than 0.05 were considered to be statistically significant.

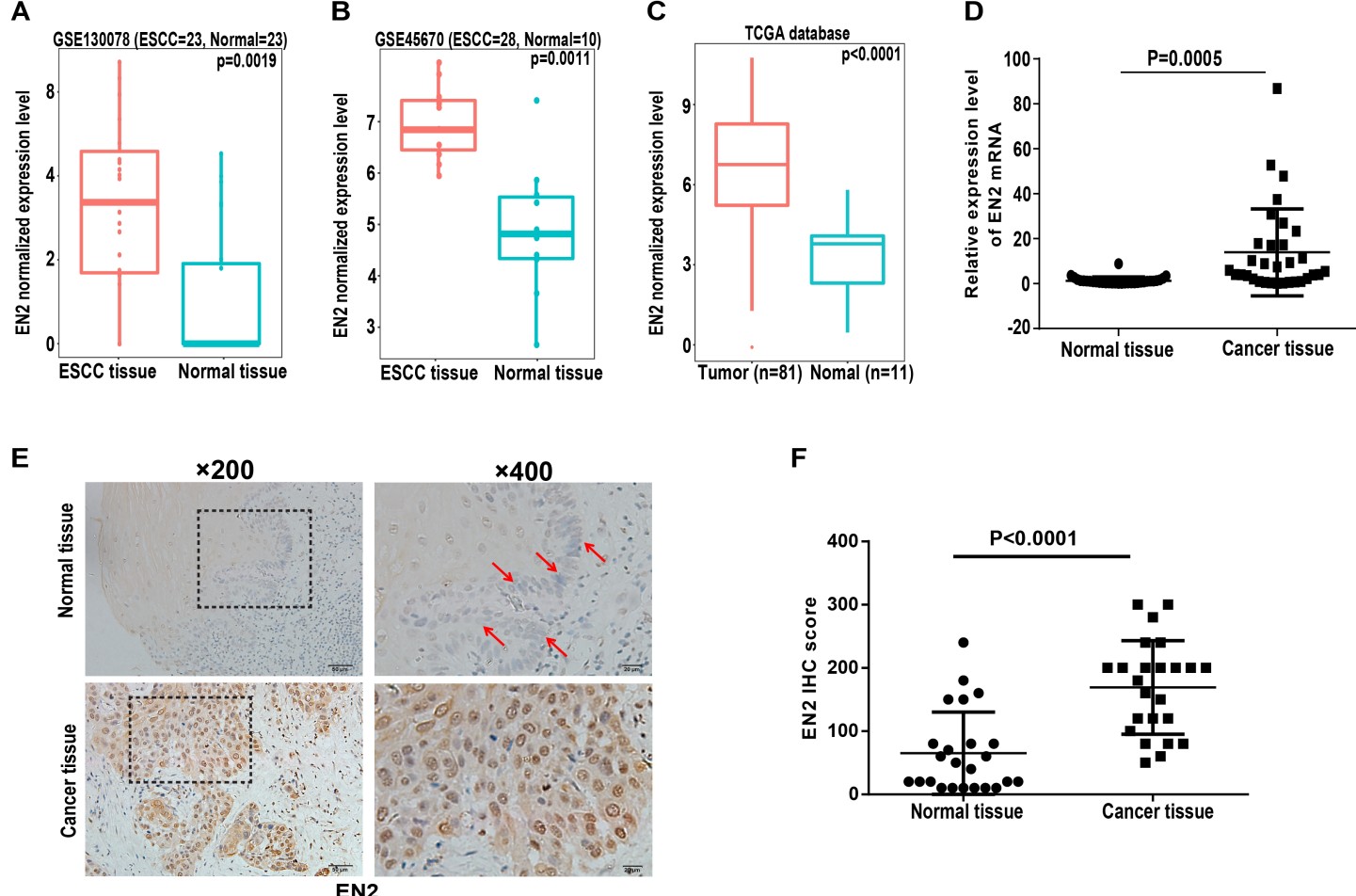

**Figure 1 Expression of EN2 is significantly elevated in ESCC.** (A and B) Microarray data retrieved from gene expression omnibus (GEO) repository were utilized to analyze EN2 expression at mRNA level in ESCC tissues including GSE130078 (normal = 23, cancer = 23) and GSE45670 (normal = 10, cancer = 28). (C) Data in TCGA database showed higher mRNA level of EN2 in ESCC tissues compared to adjacent normal tissues. (D) Expression of EN2 mRNA in 32 pairs of tumor samples and adjacent normal tissues as determined using qRT-PCR. (E) Representative images of IHC staining of EN2 in ESCC and adjacent normal tissues. (F) Expression of EN2 protein as determined by IHC (*n* = 24). The IHC score of EN2 was calculated as the staining intensity (0, 1, 2, or 3) × positive percentage × 100.

## RESULTS

### EN2 is highly expressed in ESCC tissues

Engrailed-2, a member of the homeobox superfamily, is highly expressed in a number of cancers, including cancers of the bladder, prostate, and breast (*Bose, Bullard & Donald, 2008*; *Martin et al., 2005*; *Morgan et al., 2011, 2013*). The function of EN2 in ESCC is currently unknown and its role was evaluated to further reveal the molecular events linked to the carcinogenesis of ESCC. The expression level of EN2 was first bioinformatically analyzed using two individual datasets from the NCBI GEO database, including GSE130078 and GSE45670 as well as on TCGA database. The mRNA expression of EN2 was found to be significantly elevated in ESCC tissues (Figs. 1A–1C). Surgical specimens derived from ESCC patients were used to examine the expression level of EN2 using

quantitative real-time polymerase chain reaction (qRT-PCR) to support these results. The mRNA expression levels of EN2 were significantly upregulated in ESCC tissues compared with the paired adjacent normal tissues ($p = 0.0005$; Fig. 1D). Immunohistochemistry was used to analyze the expression of the EN2 protein in tissues. The EN2 protein expression in ESCC was significantly elevated compared with the adjacent normal tissues (Figs. 1E and 1F). These results confirmed that EN2 is highly expressed in ESCC at both the mRNA and protein-specific levels.

## EN2 promotes proliferation and clonogenic abilities of ESCC cell lines

Homeobox genes have emerged as key players in the development of cancer so understanding the function of EN2 in ESCC is of great interest. The mRNA expression levels of EN2 were examined in three ESCC cell lines, namely Eca109, Kyse150 and TE-1, as well as the SHEE cell line, which is an immortalized epithelial cell line derived from the fetal esophageal epithelium induced by HPV 18 E6E7 AAV (*Shen et al., 2003*). A normal RNA sample was included as a reference, which had an equal molar ratio of total RNA samples from the 32 normal samples to that of the tumor samples from the 32 ESCC patients. The mRNA expression of EN2 in Eca109 and Kyse150 was lower than TE-1 (Fig. 2A), and the latter two cell lines were used to study the effect of exogenously expressed EN2 on ESCC cell lines. The lentiviral system was adopted to introduce EN2 into the cell lines. The expression of EN2 transcripts was greatly increased, as was expected (Fig. 2B). The expression of the EN2 protein was confirmed using an immunofluorescence assay (Figs. 2C and 2D). The growth capacity of the EN2-infected cells was evaluated and the growth rates of both EN2-Eca109 and EN2-Kyse150 were significantly increased compared with those of NEO-Eca109 and NEO-Kyse150 (Fig. 2E). Cell growth was also measured using the IncuCyte Zoom live-cell imaging assay, which showed that EN2 promoted the proliferation of the two cell lines (Fig. 2F). The effect of EN2 on the colony-forming ability of Eca109 and Kyse150 was also examined and the results revealed that both EN2-infected cell lines obtained the augmented capacity of colony formation, however, the effect was more apparent in Eca109 (Figs. 2G–2J).

## EN2 facilitates migration and invasion of ESCC cell lines

Cancer cells are characterized by their ability to migrate and invade other cells. We tried to determine the effects of EN2 on these malignancy-associated phenotypes. The wound-healing assay was used to examine the effects of EN2 on the migration of ESCC cell lines; EN2-Kyse150 (Figs. 3A and 3B) and EN2-Eca109 (Figs. 3C and 3D) cells migrated faster than did their respective empty controls. Their invasive ability was examined in the Matrigel using a live-cell imaging assay. The results revealed that both Kyse150 (Figs. 3E and 3F) and Eca109 (Figs. 3G and 3H) were more aggressive in the Matrigel after the introduction of exogenous EN2. These results indicate that EN2 could promote the malignant phenotypes of ESCC cells.

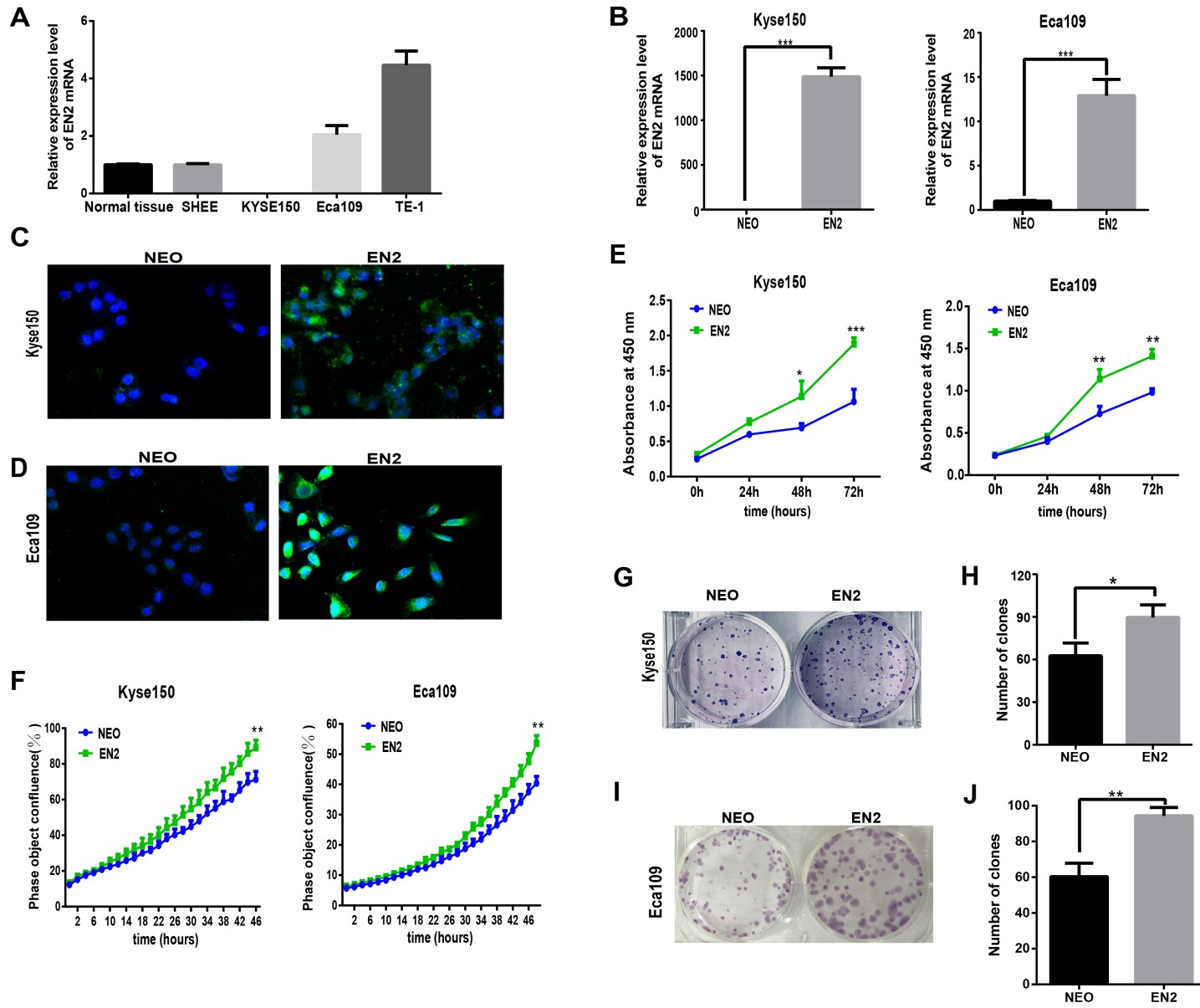

**Figure 2  EN2 expression augments the growth and clonogenic abilities of ESCC cell lines.** (A) The mRNA expression of EN2 in three ESCC cell lines, in a mixture of total RNA extracted from normal esophageal samples adjacent to ESCC tumors of 32 patients, and in the SHEE control cell line was evaluated by means of qRT-PCR. (B) Eca109 and Kyse150 cells were infected with either EN2-containing or NEO-containing empty lentivirus particles. The mRNA expression level of EN2 was evaluated using qRT-PCR. (C and D) Fluorescent images of cancer cells stained with anti-EN2 antibody. EN2 staining is shown in green using FITC-labeled secondary antibody. Cell nuclei were stained blue (DAPI) and images were taken by confocal microscopy at ×20 magnification. (E) EN2 promoted cell growth as demonstrated by CCK-8 assay. (F) EN2 promoted cell growth as determined by the IncuCyteZOOM® live cell imaging assay. (G and H) Exogenous EN2 expression augmented the colony-forming ability of Kyse150 cell lines. (I and J) Exogenous EN2 expression augmented the colony-forming ability of Eca109 cell lines (*$p < 0.05$, **$p < 0.01$, ***$p < 0.001$).

## Disruption of EN2 inhibits the proliferation, clonogenicity, migration and invasion of ESCC cells

The CRISPR/Cas9 system was used to investigate the biological function of EN2 in ESCC by generating EN2 null cells in TE-1, in which EN2 is highly expressed. The CRISPR/Cas9

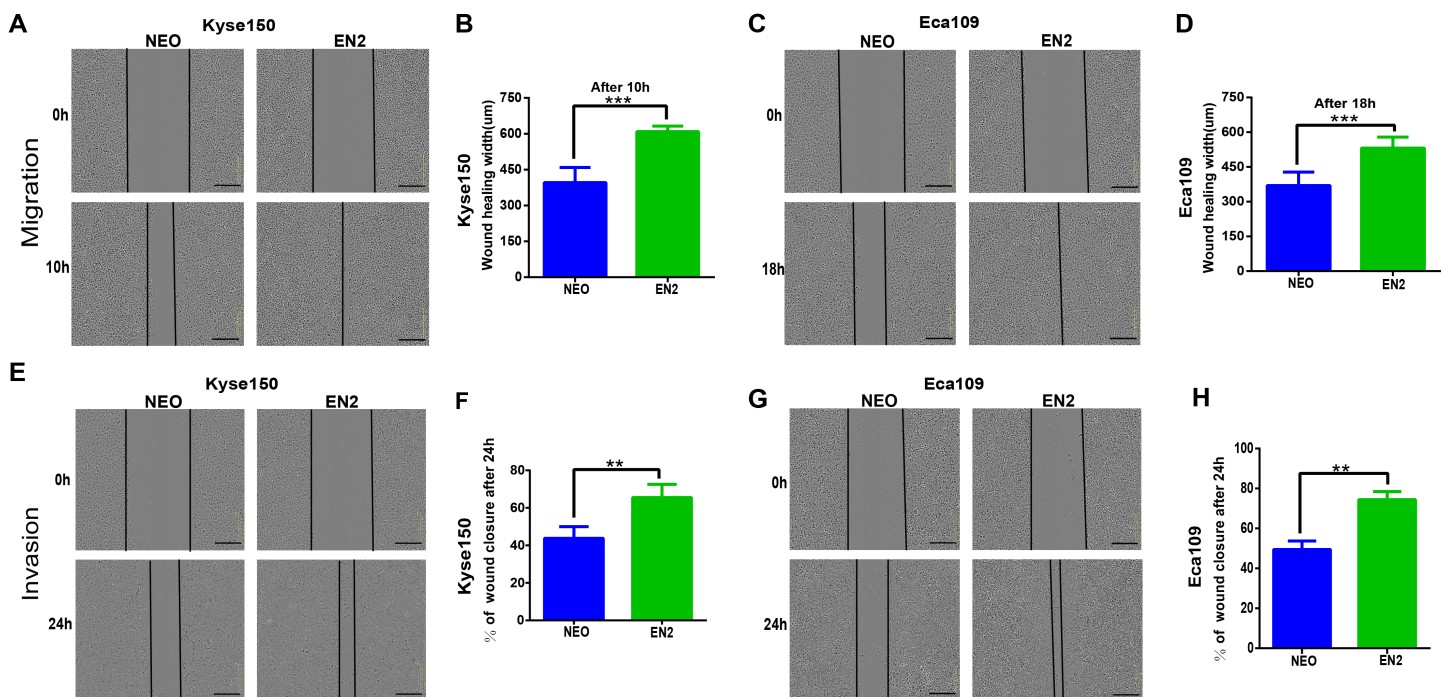

**Figure 3 EN2 promotes the migration and invasion of ESCC cells.** (A and B) Migratory variation of Kyse150 cells expressing EN2 or NEO was evaluated using wound-healing assay. (C and D) Migratory variation of Eca109 cells expressing EN2 or NEO was evaluated using wound-healing assay. (E and F) Invasive variation of Kyse150 cells expressing EN2 or NEO was evaluated using Matrigel invasion assay. (G and H) Invasive variation of Eca109 cells expressing EN2 or NEO was evaluated using Matrigel invasion assay. Representative photographs (left) and quantification (right) are shown. The scale bars represent 300 μm (**$p < 0.01$, ***$p < 0.001$).

machinery was designed to destroy the gene structure of the first exon (Fig. 4A). One positive clone was chosen for experimentation. The successful disruption of the EN2 gene was confirmed using the mismatch-sensitive T7 endonuclease assay (Fig. 4B), in which the appearance of bands of 744 bp and 354 bp indicated the expected modification of the target site in the EN2 locus. The PX458 empty vector was transfected into TE-1 cells to generate a control cell line. The expression of the EN2 protein was eliminated in the EN2-disrupted cells as determined by the immunofluorescent staining assay (Fig. 4C). We subsequently assessed whether the disruption of EN2 affects the malignant phenotype of TE-1 cells. TE-1 cells with the disrupted EN2 locus exhibited a reduced growth rate (Fig. 4D). Their colony-forming ability was also weakened when compared with the control (Figs. 4E and 4F). The disruption of EN2 hindered the migration (Figs. 4G and 4H) and invasive (Figs. 4I and 4J) behavior of TE-1 cells. These results demonstrate that the disruption of EN2 negatively affects the malignant phenotype of TE-1 cells and further supports its oncogenic role in ESCC.

## EN2 upregulates the expression of SPARC

EN2 is similar to other homeodomain proteins in its ability to bind to the sequence-specific DNA sites in the chromosome and consequently regulate the downstream genes like other transcription factors. The genetic expression profiling in EN2-Eca109 and EN2-Kyse150 cells was analyzed to determine the molecular mechanisms responsible for their

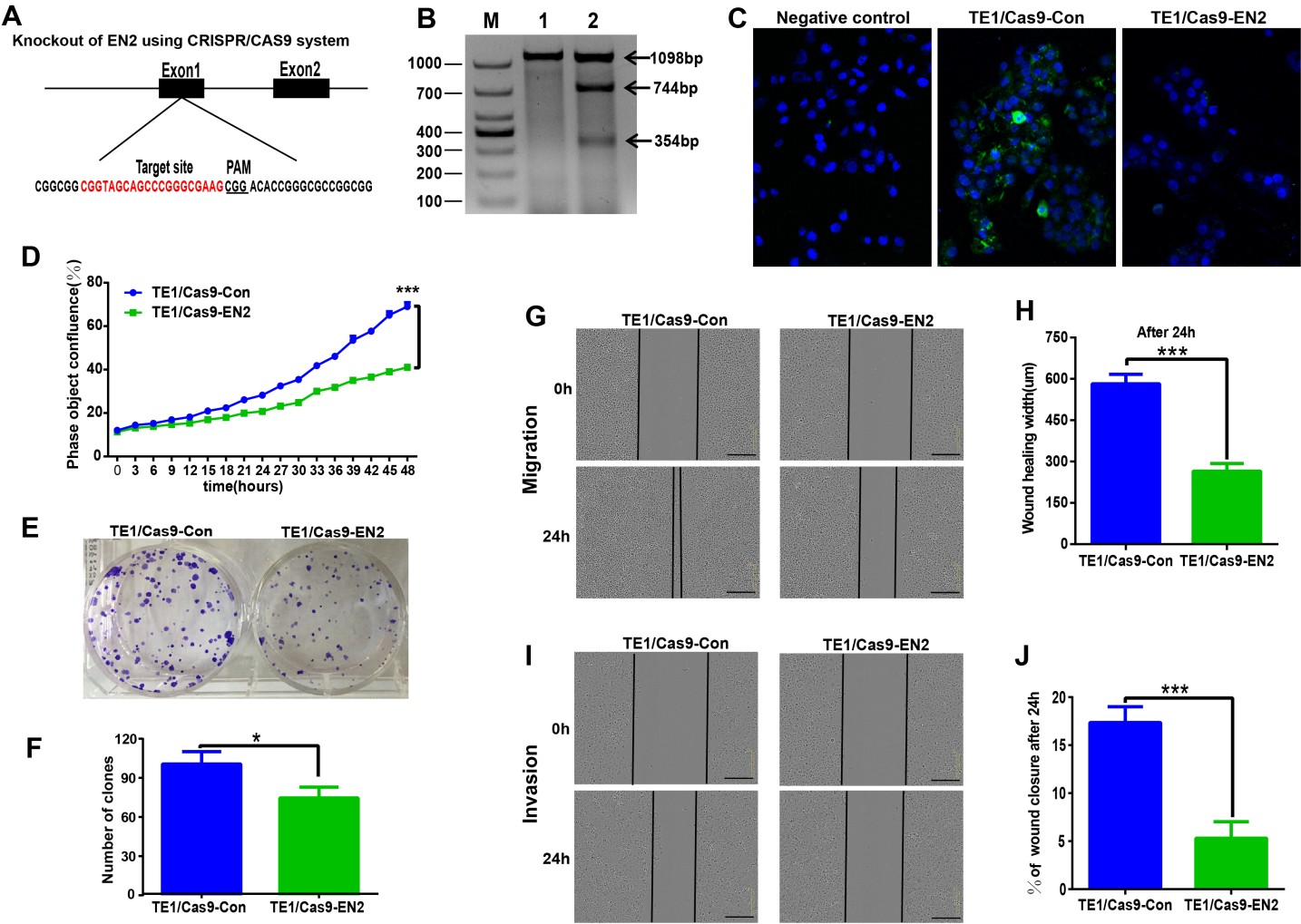

**Figure 4** **Silencing of EN2 negatively affects the proliferation, clonogenicity, migration and invasion of TE-1 cells.** (A) Schematic representation of the CRISPR-Cas9 target site (marked in red), which is located at the first exon of the EN2 locus. The protospacer adjacent motif (PAM) is underlined. (B) The confirmation of effective disruption of the EN2 gene using the T7 endonuclease assay. The size of the T7 endonuclease I-digested DNA fragments is indicated on the right; lane M, 1 and 2 indicate the DNA marker (DL1000; Takara, Dalian, China), the sample from the parental TE-1 cells transfected PX458 vector (control), and the sample from the EN2-disrupted cells. (C) The confirmation of effective disruption of EN2 protein using immunofluorescent staining. EN2 staining is shown in green (FITC-labeled secondary antibody). Cell nuclei are stained in blue (DAPI), images were taken by confocal microscopy at ×20 magnification. (D) Disruption of EN2 decreases cell growth as determined by the live-cell imaging assay. (E and F) Disruption of EN2 hampers the colony-forming ability of TE-1 cells. (G and H) Disruption of EN2 suppresses the migratory potential of TE-1 cells. (I and J) Disruption of EN2 suppresses the invasive capacity of TE-1 cells. The scale bars equal 300 μm (*$p < 0.05$, ***$p < 0.001$). 

protumor functions. The transcriptomic data were obtained using the Affymetrix clariom S array to examine their genetic expression. A total of 62 genes were found to be upregulated (Table S2) and 42 genes were downregulated (Table S3) in the EN2-Eca109 cells as compared with the NEO-Eca109 cells (Fig. 5A). A total of 131 genes were found to be upregulated (Table S4) and 249 genes downregulated (Table S5) in the EN2-Kyse150 cells compared with the NEO-Kyse150 cells (Fig. S1A).

Gene ontology analysis was conducted to determine the biological function of EN2-regulated genes. The genes upregulated by EN2 in Eca109 are categorized into groups

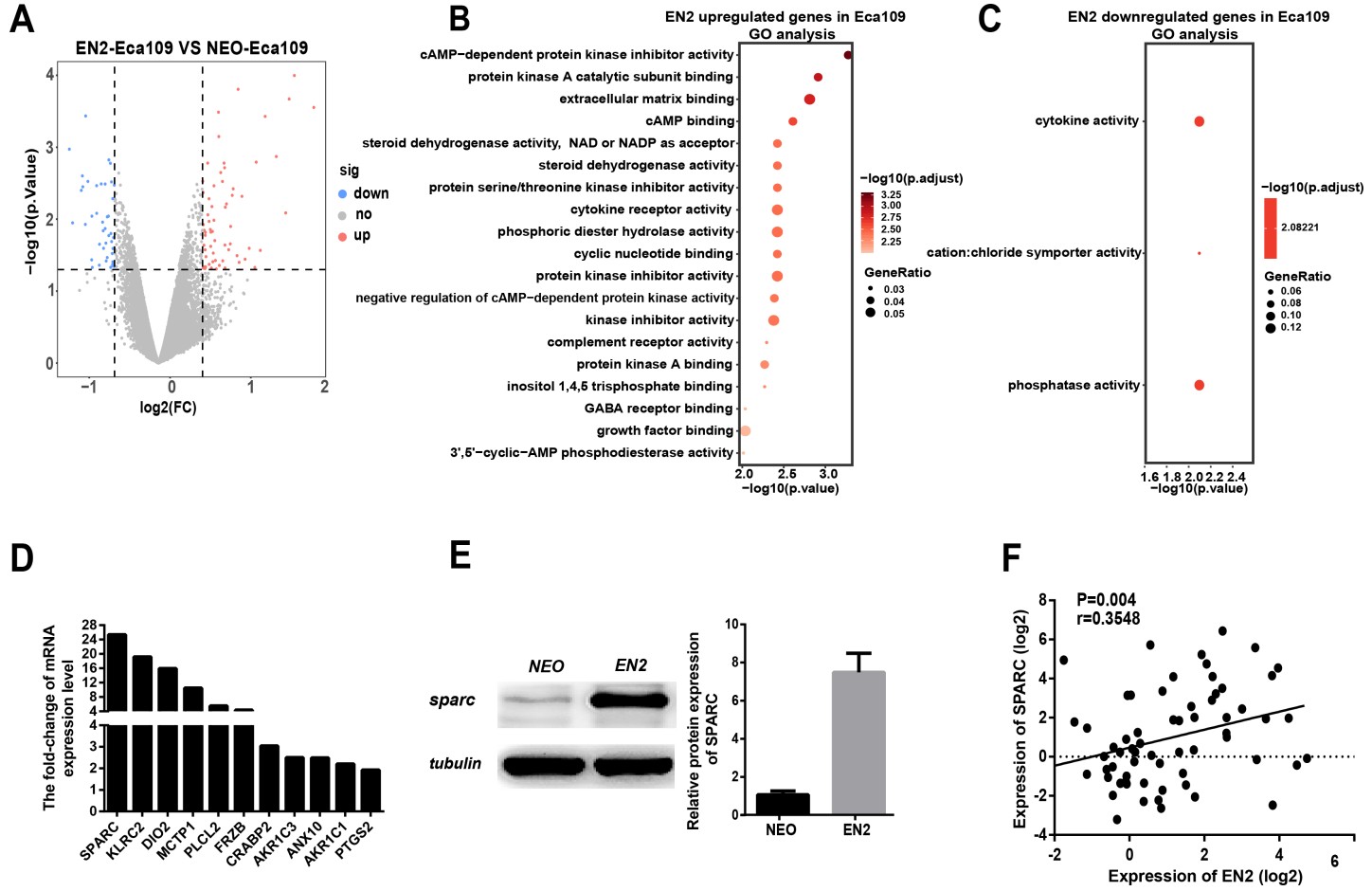

**Figure 5 EN2 upregulates the expression of SPARC.** (A) Volcano maps show the differentially expressed genes in EN2-Eca109 as compared with the NEO-Eca109 cells. (B) Gene ontology (GO) analysis identified biological processes impacted by EN2 upregulated genes in Eca109 cells. (C) Gene ontology (GO) analysis identified biological processes impacted by EN2 downregulated genes in Eca109 cells. (D) Expression of the upregulated genes with high fold-change in EN2-Eca109 and NEO-Eca109 cells by means of qRT-PCR. (E) Examination of SPARC protein in EN2-Eca109 and NEO-Eca109 cells using the western blot assay. (F) Correlation analysis of EN2 and SPARC expression in ESCC tissue samples (*n* = 32).

including cAMP-dependent kinase inhibition, extracellular matrix binding, protein kinase A activity, growth receptor binding, cytokine receptor activity, and phosphoric diester hydrolase activity (Fig. 5B). The genes upregulated by EN2 in Kyse150 are categorized into groups including ErbB-2 class receptor binding, extracellular matrix constituent, proteinaceous extracellular matrix, and response to type I interferon (Fig. S1B). The results of the GO analysis of the downregulated genes are shown in Fig. 5C and Fig. S1C respectively. These data suggest that EN2 is able to upregulate genes connected to the malignant characteristics of ESCC cells. A segment of gene expression was verified using qRT-PCR analysis (Fig. 5D; Fig. S1D). There are many well-documented pro-oncogenic genes (*Kojima et al., 2019*; *Tian et al., 2016*; *Wu et al., 2019*) that are upregulated upon the expression of EN2, implying that EN2 could modulate the expression of the downstream targets genes in a cellular, context-dependent manner.
The upregulated oncogenic gene was investigated for its role in mediating the biological function of EN2 using SPARC in the Eca109 cell line. The secreted protein acidic and rich in cysteine (SPARC) is a multi-faceted protein that has a well-documented function in the malignant features of various types of cancer (*Chen et al., 2012*; *Hung et al., 2017*). The mRNA level of the SPARC gene increased by about 25 fold with EN2 in Eca109 cells (Fig. 5D). The protein level also increased by approximately 7.5 fold, as determined by western blot testing (Fig. 5E). Our results indicate a drastic induction of SPARC in EN2-Eca109, implying a possible positive correlation between the expression of EN2 and SPARC. We examined both EN2 and SPARC expression in the clinical ESCC samples using qRT-PCR; the results verified the positive correlation of ESCC (Fig. 5F). The expression of SPARC modulated by EN2 was investigated in two other cell lines. The mRNA expression level of SPARC was found to decrease by about 60% in EN2-null TE-1 cells as determined by qRT-PCR. However, EN2 did not change the expression of SPARC in EN2-Kyse150 cells when compared with its expression in NEO-Kyse150 cells (Fig. S1E). These data demonstrate that EN2 is able to boost the expression of the SPARC genes in Eca109 and TE-1 but not in Kyse150.

## The homeodomain is essential for the SPARC induction and the protumor function of EN2

The sequence and structure of the homeodomain are highly conserved in the members of the homeobox superfamily. The homeodomain of EN2 has a three-helix structure that directly participates in DNA binding. Two mutants were created to evaluate the significance of the homeodomain in the protumor function of EN2. The mutants each had a small deletion in the homeodomain; the deletions were as follows: mut1 with a deletion of the residues from 283 to 292 and mut 2 with a deletion of the residues from 293 to 302 (Fig. 6A). The sequencing maps around the mutated sites are shown in Fig. 6B. The expression vectors, each containing a mutated EN2, were introduced respectively into Eca109 using the lentivirus system and generating the Mut.1-Eca109 and Mut.2-Eca109 cell lines. The expression of the two mutant transcripts of EN2 was verified using qRT-PCR (Fig. 6C). The effects of the two mutants were evaluated for their ability to proliferate, migrate and invade the cell line. Both mutants were shown to hinder the promoting effects of EN2 on the proliferation of Eca109 (Fig. 6D). The mutants largely eliminated the enhancing effects of EN2 on the cell's migratory abilities (Figs. 6E and 6F) and invasion in matrigel (Figs. 6G and 6H). The expression status of SPARC in the Mut1-Eca109 and Mut2-Eca109 cell lines was investigated and the results revealed that the mutations in the homeodomain also abrogate the function of EN2 in inducing the expression of SPARC (Figs. 6I and 6J). Our results indicate that the homeodomain is essential for the protumor effect of EN2.

## ShRNA-mediated knockdown of SPARC hampers the pro-oncogenic effect of EN2

SPARC may be necessary for mediating the biological functions of EN2. We generated a shRNA lentiviral vector specifically directed against the SPARC transcript to determine

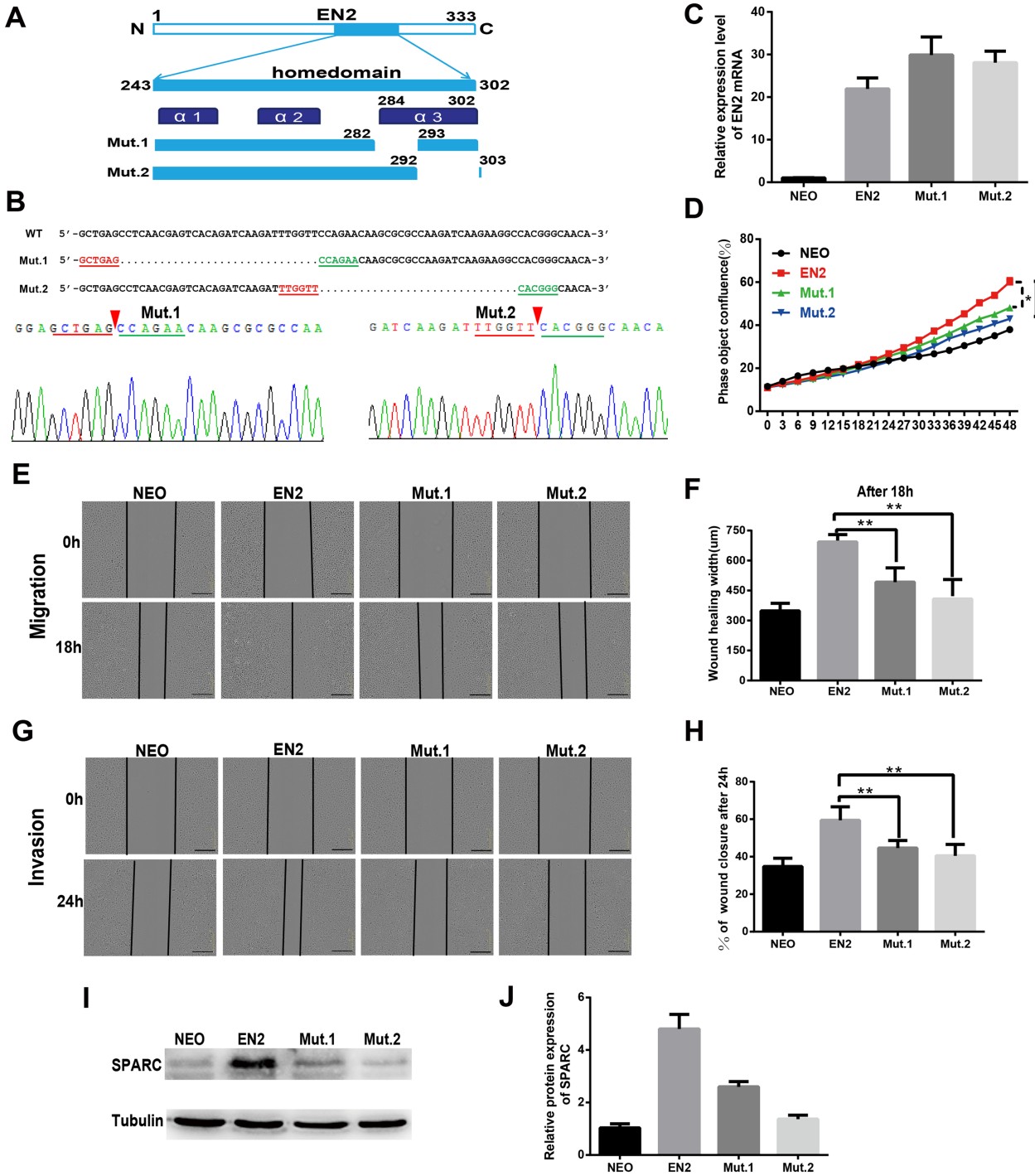

**Figure 6 The homeodomain is essential for the SPARC induction and the protumor function of EN2.** (A) Schematic map of the two EN2 mutants with small deletions in the homeodomain region. α1, α2 and α3 indicate the three-helix structure in the homeodomain. (B) Confirmation of the expected mutations using the Sanger sequencing assay. The red triangles indicate the positions where part of the sequences are deleted. (C) The mRNA expressions of the EN2 and mutants were analyzed by qRT-PCR in Eca109 cells. EN2, Mut.1 and Mut.2 indicate the Eca109 cells infected with wild-type EN2, mutant 1 and mutant 2 expression lentivirus, respectively. (D) Analysis of the effect of the two mutants of EN2 on cell proliferation. (E and F) Analysis of the effect of the two mutants of EN2 on cell migration. (G and H) Analysis of the effect of the two mutants of EN2 on cell invasion. The scale bars represent 300 μm. (I and J) The mutations in the homeodomain attenuate the SPARC-inducing ability of the wild-type EN2 as determined by western blot assay (*$p < 0.05$; **$p < 0.01$).  

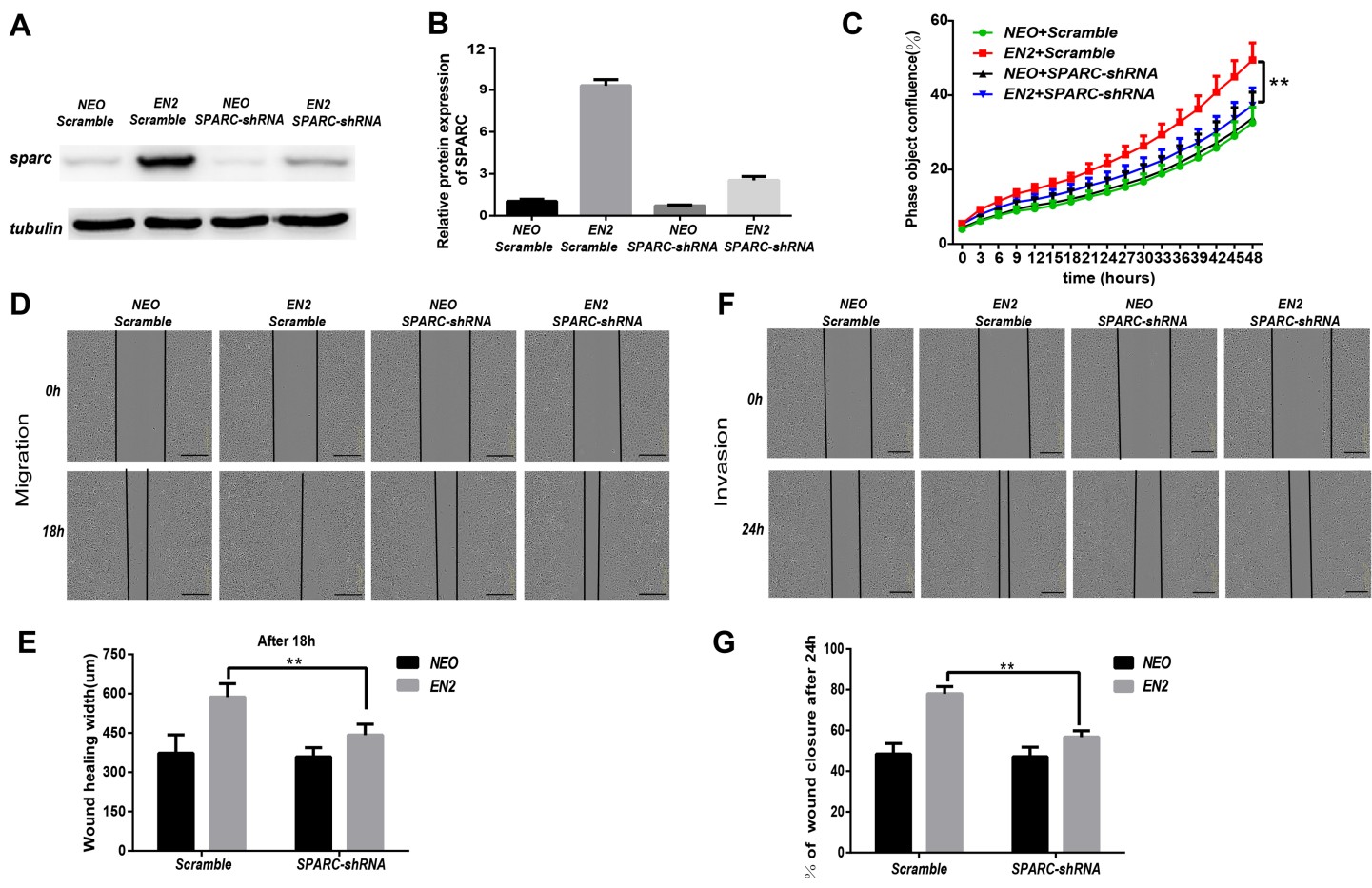

**Figure 7 ShRNA-mediated knockdown of SPARC hampers the pro-oncogenic effect of EN2.** (A and B) Analysis of the efficiency of SPARC knockdown in EN2-Eca109 cells by means of the western blot assay. (C) The effect of SPARC knockdown on EN2-induced cell proliferation as determined by the live-cell imaging assay. (D and E) The effect of SPARC knockdown on EN2-induced cell motility as determined by the wound healing assay. (F and G) The effect of SPARC knockdown on EN2-induced cell invasion as determined by the Matrigel invasion assay. The scale bars represent 300 μm ($^{**}p < 0.01$).

whether the EN2-mediated tumor proliferation, migration and invasion are dependent on SPARC expression. A scramble sequence was used as the control. The introduction of the shRNA vector into NEO-Eca109 and EN2-Eca109 successfully reduced the expression of SPARC proteins by 80% (Figs. 7A and 7B). Knockdown of SPARC significantly reduced the proliferation capacity of EN2-Eca109 (Fig. 7C). We also observed the decreased capacities of the migration and invasion of the shRNA-transfected cells (Figs. 7D–7G). This result demonstrates that SPARC plays a critical role in mediating the biological function of EN2 in the Eca109 cell line.

## DISCUSSION

The poor prognosis and high mortality rates of esophageal cancer highlight the need to better understand the molecular events leading to the progression of this disease. We demonstrated that engrailed-2, a member of the homeobox superfamily, may be a

potential biomarker in the identification of the malignancy of ESCC and confirmed its oncogenic role in this cancer.

Certain members of the homeobox superfamily have been implicated in normal development as well as in the pathological development of cancer. OCT-4 is a homeobox gene well known for its function in generating induced pluripotent stem cells (*Bhartiya, 2013*; *Seiler et al., 2011*); this gene has also been identified as an oncogene known for promoting the features of cancer stem cells (*Li et al., 2017*; *Phiboonchaiyanan & Chanvorachote, 2017*). In addition to OCT-4, an increasing body of evidence reveals that homeobox genes may participate in the promotion of various malignant phenotypes such as the proliferation, migration and invasion of cancers (*Hirao et al., 2019*; *Miwa & Kanda, 2019*; *Rodini et al., 2012*; *Song et al., 2019*). We have previously reported that HoxC6, another member of homeobox superfamily, functions as an oncogene in ESCC (*Tang et al., 2019*). We therefore performed the correlation analysis using the data retrieved from TCGA, and the result indicates that these two genes function independently (Fig. S2). Nevertheless, these results suggest that HoxC6 and EN2 may complement each other when using as the potential biomarkers in the diagnosis or treatment of ESCC. Additional homeobox genes will likely be recognized as critical factors in the malignancy of ESCC, which may provide clinical opportunities for the diagnosis and treatments of this cancer in the future.

EN2 functions as a transcription factor, much like other homeobox genes, binding directly to specific DNA sites in the chromatin via the conserved homeodomain and controlling the expression of downstream targets. In our experiment, EN2 was able to modulate the expressions of hundreds of downstream targets. Although the expression of distinct sets of genes are affected in various ESCC cell lines, it is likely that many of these genes are connected to the mailgnant phenotype of cancer cells. For instance, the expression of SPARC is significantly upregulated upon the expression of EN2 in two of three ESCC cell lines. We further demonstrated that SPARC is capable of mediating the oncogenic effect of EN2 in the Eca109 cell lines. A similar effect was not observed for EN2 on the expression of SPARC in Kyse150 cell lines; however, there were many other oncogenic genes regulated upon the expression of EN2 in this cell line. This phenomenon suggests that the mechanism underlying the biological function of EN2 may be cell context-dependent. Our work confirms the importance of the homeodomain on the pro-malignant effect of EN2 in ESCC cells. However, the homeodomain alone is not sufficient to determine the DNA-binding specificity of a homeobox gene-encoded protein and the involvement of the amino acid sequence outside of the homeodomain region is also required (*Van Dijk & Murre, 1994*; *Peltenburg & Murre, 1997*). The availability of other co-factors and the accessibility of chromatin also influences the complexity of this system (*Porcelli et al., 2019*), and may influence the outcome of EN2 expression in various cell lines. Further insight into the molecular mechanisms will help to explain the biological effects of EN2 in ESCC.

The data presented in this work demonstrates that the biological effect of EN2 is correlated with its role in transcription activation. Future studies should seek to clarify the structural basis and components of EN2-containing transcription activation complexes.

## CONCLUSION

We demonstrated that EN2 plays an oncogenic role in ESCC. Further, the biological function of EN2 is mediated via the upregulation of downstream oncogenes. Our work provides more information on the molecular mechanisms underlying the malignancy of this type of cancer. EN2 may serve as a potential candidate as a diagnostic marker or therapeutic target of ESCC in the future.

## ACKNOWLEDGEMENTS

We would like to thank our colleagues for their contribution to this study.

### Funding

This study was supported by grants from the Joint Program on the Science and Technology Collaboration of Southwest Medical University and the Government of Luzhou City (2018LZXNYD-PT04), the Science Technology Support Plan Projects of Luzhou (2015LZCYD-S02), and the Program of Southwest Medical University (2017-ZRQN-005). The funders had no role in study design, data collection and analysis, decision to publish, or preparation of the manuscript.

### Grant Disclosures

The following grant information was disclosed by the authors:
Science and Technology Collaboration of Southwest Medical University and the Government of Luzhou City: 2018LZXNYD-PT04.
Science Technology Support Plan Projects of Luzhou: 2015LZCYD-S02.
Southwest Medical University: 2017-ZRQN-005.

### Competing Interests

The authors declare that they have no competing interests.

### Author Contributions

- Yong Cao performed the experiments, analyzed the data, prepared figures and/or tables, authored or reviewed drafts of the paper, and approved the final draft.
- Xiaoyan Wang performed the experiments, prepared figures and/or tables, authored or reviewed drafts of the paper, and approved the final draft.
- Li Tang performed the experiments, authored or reviewed drafts of the paper, and approved the final draft.
- Yan Li performed the experiments, authored or reviewed drafts of the paper, and approved the final draft.
- Xueqin Song performed the experiments, prepared figures and/or tables, and approved the final draft.
- Xu Liu analyzed the data, prepared figures and/or tables, and approved the final draft.
- Mingying Li performed the experiments, prepared figures and/or tables, and approved the final draft.
- Feng Chen performed the experiments, prepared figures and/or tables, and approved the final draft.
- Haisu Wan conceived and designed the experiments, authored or reviewed drafts of the paper, and approved the final draft.

## Human Ethics

The following information was supplied relating to ethical approvals (i.e., approving body and any reference numbers):

This study was approved by the ethics review board at the Affiliated Hospital of Southwest Medical University (K2018002-R).

## Data Availability

Data is available at GEO: GSE136331.

## Supplemental Information

Supplemental information for this article can be found online at http://dx.doi.org/10.7717/peerj.8662#supplemental-information.

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
