# Peer review of "Engrailed-2 promotes a malignant phenotype of esophageal squamous cell carcinoma through upregulating the expression of pro-oncogenic genes"

_PeerJ, doi:10.7717/peerj.8662_

## Round 0.1 · original submission · Major Revisions

· Academic Editor

Major Revisions

All comments of the reviewers need to be addressed in full, however, regarding the major points 1 and 3 raised by Reviewer #1, I want to suggest to at least demonstrate SPARC expression in the second cell line (Figure 5E).

In support and in addition to the comments of the Reviewers, I want to stress the following points:

(1) The manuscript must be improved concerning the English language (including the correct usage of articles), the consistent use of either past tense or present tense, and the correction of typos. Also make sure to correctly use the terms "transfection" and "transduction": infection with lentivirus is called transduction, not transfection

(2) The discussion must be revised in that the results are not just repeated but discussed and put in the context of the literature.

(3) Appropriate articles should be cited; e.g. line 42: Jiang et al. cited for clinical characteristics, but the article is about method development; or line 56: 2 papers cited for roles of homeobox genes in embryonic development and cell identity, but both articles deal with cancer.

(4) Materials and Methods:
- provide primer sequences for GAPDH in Table 1
- provide resource of all vectors, EN2 cDNA, 293FT cells
- indicate which packaging plasmids were used for lentivirus production
- line 118: is control with primary antibody correct? Or secondary antibody?
- line 125: correct citation of vectors from Addgene (see Addgene webpage)
- CRISPR paragraph: which control was used?
- line 139: only cite the most appropriate paper
- line 151: "proper concentration": please be more precise and provide range
- line 155: how many cells were seeded?
- line 164: what exactly is meant by "wound-maker"?
- line 167: provide company for Matrigel and type of Matrigel
- line 181: provide more detailed description of GO enrichment analysis
- Western blotting: what was used for blocking and dilution of antibodies?
- line 201: "pathologists were invited ..." -> did they also perform both the independent analysis or were they only invited?

(5) Figures:
- Applies to all panels that show statistics/bar charts/curves etc.: provide for each panel in the legend the number of biological replicates and the type of error bars that are shown (STD? SEM? Median? Mean? etc.)
- Fig. 1B: provide magnification
- In all panels showing multiple images (e.g. representative image and bar chart) always describe the subpanels in the legend. For example: left panel shows ..., right panel shows...
- Fig. 5A: differentially expressed genes in EN2-Eca109 relative to what?
- Fig. 5B: provide more detailed legend. Downregulated genes in what? Font size needs to be increased in image.
- Fig. 5C legends needs to be written more clearly. Font size needs to be increased in image.

(6) Additional points:
- line 211-215: do not discuss results that are not shown, or show the results. This entire part needs to be revised to make clear, why EN2 was studied.
- line 221: "... ESCC was significantly elevated ..." -> how many samples were studied to draw this conclusion? Please show statistics in the figure. Also, how was positivity in IHC determined/quantified?
- line 228: "... was relatively low ..." -> compared to what? For my understanding only KYSE150 has a lower EN2 expression compare to controls.
- line 226: "Two cell lines, ..." -> four cell lines are shown in Figure 1A; please describe better. What is SHEE? What exactly is the normal tissue sample?
- line 238: cite Figure 2F
- line 247: cite Figure 3A
- line 259: what is meant by "disrupted EN2 group"? Please also make clear that one clone was used and what exactly the control is.
- line 271: provide the complete gene list as supplemental material
- lines 274 and 277: replace "items" by "categories"
- line 279: conclusion too general
- lines 282 ff.: briefly mention what SPARC is
- line 301-302: describe results separately
- results in general: avoid too detailed repetition of methods, only where it is really necessary.

Reviewer 1 ·

Basic reporting

English has to be revised, the authors were recommended to seek language support.

Sufficient literature references.

Data are well presented.

Experimental design

The adopted experimental design is usual for such (basic) investigations. Figures are self-explanatory; methods sufficiently described in the Material&Methods section.

Validity of the findings

The findings represent an incremental benefit to the current knowledge. The authors were suggested to improve the Discussion section, as this is just a mere repetition of the results.

Additional comments

This study involves the investigation of engrailed 2 and its downstream genes during the progression of ESCC. Using lentiviral-mediated transduction to overexpress or CRISPR/Cas9 to delete EN2, the authors describe the role of this gene in proliferation, migration and invasion of ESCC cells. Furthermore, they characterize the role of SPARC as a downstream effector for EN2. The study brings incremental knowledge to the current literature, data is well presented. The authors should seek a native English speaker to correct through the text whenever necessary; some suggestions are offered below.
Major concerns
1. The authors indicate that they focus in their study on two cancer cell lines i.e. Eca109 and Kyse109, yet they show in Fig. 5, 6 and 7 experiments only with Eca109 cells. For consistency of data, they should show, at least some of these results, whether they could confirm their results in Kyse109 cells.
2. In Fig.5E one would welcome showing taller blots (more above and below the bands) and not only the bands themselves.
3. In Fig. 7, the authors show that SPARC abrogation hampers the pro-oncogenic effect of EN2 suggesting SPRC downstream of EN2. To substantiate these, the authors should investigate whether they can restore the migration / invasion phenotype upon ectopic expression of SPARC after knocking down EN2; with other words to restore the phenotype in Figure 4 upon overexpression of SPARC.
4. The discussion chapter is a mere repetition of the results. The authors should discuss their findings with respect to what is known so far, provide eventually an outlook to the field.

Figure legends
Figure 2D – please rephrase, CCK-8 assay is just a read-out for cell growth; “….as demonstrated by CCK-8 assay”
Lane: 280-281: please provide reference.
English language
Lane 75: should read “All patients…”
Lane 215: should read “Here we sought to…”
Lane 217-218:please rephrase
Lane: 222: should read “EN2 is highly expressed in ESCC…”
Lane 226: two cell lines used….in the text / three cell lines… in Figure legend / four cell lines presented in Fig. 2A
Lane 243: “migrated faster…” instead of “much more quickly”
Lane 256-257:please rephrase
Lane 258: do you mean “assessed”?
Lane 284: should read “a drastic…”
Lane 286: rephrase
Lane 287: “These data demonstrate…”
Lane 296: do you mean “containing a mutated EN2…”
Lane 297: should read “two mutant transcripts…”
Lane 298: please rephrase, comprehension difficult
Lane 303: do you mean “assessed”?

Reviewer 2 ·

Basic reporting

This is an interesting study reporting the implication of engrailed-2 (EN2) in the tumorigenesis of esophageal squamous cell carcinoma (ESCC). Overall, the experiments are appropriate and the conclusions seem to be supported by the evidence provided. It is unclear, nevertheless, the rationale for focusing this study on EN2. Other homeobox genes were also found associated with malignant transformation in several tumor types. Thus, it should be clarified further why this study was centered around EN2 and not another homeobox gene(s).

The authors published recently a very similar paper in this journal focusing on HOXC6 in ESCC (https://doi.org/10.7717/peerj.6607). However, there is no mention to their previous study in this manuscript. The authors found similar implications of HOXC6 in the promotion of ESCC cell migration, invasion, proliferation, the activation of oncogenic genes, and claimed that “HOXC6 may be a new significant biomarker for diagnosis, therapy, and prognosis”. Therefore, the results observed for EN2 should be compared with the ones for HOXC6 to address if there is a dependency or if they are completely independent biomarkers.

The microarray raw data performed in this study should be deposited in a public repository, such as GEO or ArrayExpress, in compliance of the Data Sharing policy.

The figures of this study were provided as bitmaps instead of vectors and were displayed substantially pixelated. It would be recommended to provide vector images for graphs and figures other than pictures.

Overall, the writing of the manuscript is clear, although it should be double checked to avoid certain typos, such as ‘identity(Idaikkadar’ on line 56, ‘lentivrial’ on line 228, ‘an drastic’ on line 284 and ‘homoedomain’ on line 302.

Experimental design

The experiments performed in this work seem to be adequate. The pictures of the colony formation assay, however, are suboptimal and show a strong variation in terms of perspective and shadows. It is not indicated if colonies were counted manually or using a software like ImageJ. In case of the latter, the variation in perspective and shadows from picture to picture might represent a confounder.

The unpublished data described in lines 211-212 regarding the evaluation of a panel of homeobox genes in ESCC should be incorporated in the Supplementary section.

Validity of the findings

A major concern of this study is that findings were reported in only a few cell lines and 32 ESCC tumor samples. It would be necessary to validate these findings in a larger and independent dataset, such as The Cancer Genome Atlas (TCGA) or the International Cancer Genome Consortium (ICGC).

Additional comments

Table 1 is not essential to follow the narrative and could be moved to the Supplementary section.

---

## Round 0.2 · Minor Revisions

· Academic Editor

Minor Revisions

Thanks for addressing all comments and suggestions raised by the Reviewers and myself. The manuscript has substantially improved.

To be able to accept the manuscript, I want to ask you to implement the following remaining points:

1. Thanks for making the effort to send the manuscript to a company for language editing, which improved the overall readability. Unfortunately, the title worsened. I want to suggest the following title: "Engrailed-2 promotes a malignant phenotype of esophageal squamous cell carcinoma through upregulating the expression of pro-oncogenic genes". In addition, please correct the following remaining language issues:
- line 94: change sentence into: "EN2 may potentially act as a diagnostic marker or therapeutic target for ESCC treatment in the future."
- line 510: " ... cells were seeded in 96-well culture plates..." ("in" is missing)

2. Figure 1E: include the number of analyzed samples in the figure legend, which would be n=24 pairs as indicated in your response letter.

3. Figure 2A: the column with the normal samples has been removed, although it is still mentioned in the manuscript text and legend. Please correct. Also, please write the legend as follows: " The mRNA expression of EN2 in three ESCC cell lines, in a mixture of total RNA extracted from normal esophageal samples adjacent to ESCC tumors of 32 patients, and in the SHEE control cell line was evaluated by means of qRT-PCR."

4. As indicated by Reviewer #2, Figure 2G was replaced by a better image, but the corresponding quantification graph in 2H is still the same. Please address this point.

5. Lines 1133-1135: the supplemental tables listing the deregulated genes should be numbered and cited in the text.

6. Line 1137: correct typo "upregulted"

Reviewer 1 ·

Basic reporting

None at this stage.

Experimental design

OK.

Validity of the findings

OK.

Additional comments

None at this stage.

Reviewer 2 ·

Basic reporting

The authors fulfilled many of the suggestions requested by this reviewer, such as depositing the microarray data in a public database, showing clearer colony formation assays and improving the English, among others. However, the authors failed to provide a rationale for focusing this study on EN2. An explanation based on a bioinformatics analysis of TCGA data is provided in the rebuttal letter, but not in the manuscript. Thus, the authors should provide this information in the manuscript, including the corresponding figure of the TCGA analysis. In addition, the authors should include the figure of the lack of correlation between EN2 and HOXC6 mRNA expression as supplementary figure and provide further insight about their potential use as biomarkers in esophageal squamous cell carcinoma despite not being correlated.

Experimental design

It is nice that the authors repeated the colony formation assay to show better pictures for the Kyse150 cells in Fig. 2G, although this was not necessary. However, it is concerning that the figure of the quantification for the new colony formation experiments (Fig. 2H) is exactly the same as in the previous experiments.

Validity of the findings

A major concern of this study was that findings were reported in only a few cell lines and 32 ESCC tumor samples. The authors did not show a validation of their findings in a larger and independent dataset, such as The Cancer Genome Atlas (TCGA) as requested, even though they claim to have done some of such bioinformatics analyses from TCGA. Therefore, the authors should provide TCGA analyses to support their findings in the manuscript.

---

## Round 0.3 · accepted · Accept

· Academic Editor

Accept

Thanks for implementing all the requested changes, which greatly improved the manuscript. I am very pleased that we can now accept your work for publication.